# PI3Kδ activates E2F1 synthesis in response to mRNA translation stress

Sivakumar Vadivel Gnanasundram [1], Slovénie Pyndiah[1], Chrysoula Daskalogianni[1], Kate Armfield[2], Karin Nylander[3], Joanna B. Wilson[2] & Robin Fåhraeus[1,3,4]

The *c-myc* oncogene stimulates ribosomal biogenesis and protein synthesis to promote cellular growth. However, the pathway by which cells sense and restore dysfunctional mRNA translation and how this is linked to cell proliferation and growth is not known. We here show that mRNA translation stress in cis triggered by the gly-ala repeat sequence of Epstein–Barr virus (EBV)-encoded EBNA1, results in PI3Kδ-dependent induction of *E2F1* mRNA translation with the consequent activation of c-Myc and cell proliferation. Treatment with a specific PI3Kδ inhibitor Idelalisib (CAL-101) suppresses E2F1 and c-Myc levels and causes cell death in EBNA1-induced B cell lymphomas. Suppression of PI3Kδ prevents E2F1 activation also in non-EBV-infected cells. These data illustrate an mRNA translation stress–response pathway for E2F1 activation that is exploited by EBV to promote cell growth and proliferation, offering new strategies to treat EBV-carrying cancers.

[1] Inserm UMRS1162, Equipe Labellisée la Ligue Contre le Cancer, Institut de Génétique Moléculaire, Université Paris 7, Hôpital St. Louis, 75010 Paris, France. [2] School of Life Sciences, College of Medical, Veterinary and Life Sciences, University of Glasgow, Glasgow G12 8QQ, UK. [3] Department of Medical Biosciences, Umeå University, Building 6M, SE-901 85 Umeå, Sweden. [4] RECAMO, Masaryk Memorial Cancer Institute, Zluty kopec 7, 65653 Brno, Czech Republic. Sivakumar Vadivel Gnanasundram and Slovénie Pyndiah contributed equally to this work. Correspondence and requests for materials should be addressed to R.Fåh. (email: robin.fahraeus@inserm.fr)

Cellular growth and proliferation is coordinated by E2F transcription factors via multiple downstream target genes, including *cyclins* and *c-myc*[1,2]. The E2Fs are retained in an inactive state by binding to the retinoblastoma protein (pRb), p130 and p107 pocket proteins[3]. The phosphorylation of pRb by cell cycle kinase activity releases E2Fs and triggers the exit from $G_0$ into $G_1$ and S-phase and cellular proliferation[4–6]. The pRb–E2F interaction is the target of oncogenic viruses such as the human papilloma virus (HPV), Simian virus 40 (SV40) and adenovirus that via E7, Large T and E1A, respectively, compete for E2F binding to pRb[7–9]. To date, ten E2F transcription factors have been described, with E2F1–6 binding the same DNA consensus motif. Whereas E2F1–3 are gene activators, E2F4–6 are considered suppressors[1].

Downstream of the *c-myc* oncogene are numerous target genes regulating most aspects of cellular biology, illustrating the complex homoeostasis required to coordinate cellular growth and division[10]. Stimulation of ribosomal biogenesis by c-Myc is an important aspect of cellular growth[11] but the pathways that sense ribosomal activity and how this feeds back and is integrated with growth and proliferative signalling pathways are poorly understood.

The endemic form of Burkitt's lymphoma is characterised by translocation of *c-myc* to the immunoglobulin locus and by expression of the Epstein–Barr virus (EBV)-encoded EBNA1, which is the only viral antigen expressed in this cancer[12,13]. But the role of c-Myc in controlling host cell proliferation during normal EBV infection is unknown. Two independent transgenic EBNA1 mouse models show inverse correlation between the levels of EBNA1 expression and lymphoma incidence[14,15]. The phenotype discrepancy between high and low EBNA1-expressing animals has not been explained. EBV is also tightly associated with other human cancers, most notably nasopharyngeal carcinoma and Hodgkin's lymphoma, but despite knowing the link between EBV and human cancers for over half a century, the oncogenic activities of EBV are barely understood[16].

EBNA1 carries a repeat sequence of glycines and alanines (GAr), which suppresses *EBNA1* mRNA translation in cis in order to minimise the production of EBNA1-derived antigenic peptides for the major histocompatibility (MHC) class I pathway[17,18]. The translation inhibitory capacity of the GAr in cis makes it a unique tool for studying the cellular response to dysfunctional mRNA translation[19].

The PI3Kδ belongs, together with PI3Kα and PI3Kβ, to the class IA of the PI3K family that use phosphatidylinositol (4,5) bisphosphate (PI(4,5)P$_2$) as a substrate to generate the lipid second messenger PI(3,4,5)P$_3$ following tyrosine kinase receptor activation and the recruitment of the p85-type regulatory subunits in complex with the p110 catalytic subunits[20]. The PI (3, 4, 5) P$_3$ activates AKT-dependent and AKT-independent downstream pathways that control a broad range of cell responses including proliferation, metabolism and survival. The p110α and p110β subunits are expressed ubiquitously while p110δ is predominantly expressed in leucocytes but is also detected in tumour cells of solid origin[21–23]. The activation of B cell and cytokine receptors increase (PI(3,4,5)P$_3$) levels in a p110δ-dependent fashion and p110δ plays important roles in the inflammatory and allergenic response and mutated p110δ is found in patients suffering from primary immunodeficiency[24]. The compartmentalisation and homoeostasis of the TLR4 is regulated by p110δ independently of mTOR[25]. The immune regulatory capacity of p110δ also includes T cell-mediated immune tolerance, giving it a potential broad role in cancer development[26]. Drugs such as Idelalisib (CAL-101) that specifically inhibit p110δ are used to treat chronic lymphocytic leukaemia, follicular B cell Hodgkin lymphoma and relapsed small lymphocytic lymphoma[27].

We have exploited the unique capacity of the EBNA1 to suppress its own mRNA translation in cis to show how mRNA translation stress is integrated with growth stimulatory pathways. We show that EBV, like the simian, adeno and human papilloma viruses also targets E2F1 but by using a unique mechanism that does not involve the pRb but requires the activation of the PI3Kδ and the induction of E2F1 synthesis. The data help explain the oncogenic activity of EBNA1 and suggests that in addition to suppressing the production of antigenic peptides, EBNA1-mediated mRNA translation also promotes cell growth and proliferation.

## Results

**The GAr stimulates c-Myc and cell proliferation**. The glycine–alanine (GAr) repeat of the EBV-encoded EBNA1 disrupts mRNA translation in cis and offers a unique tool to study the cell biological effects of disrupted translation of individual mRNAs that cannot be achieved using chemical compounds[17–19]. When the GAr was expressed in H1299 human carcinoma cell lines, a significant increase (~35%) in colony formation was observed when compared to plasmid control (EV) (Fig. 1a and Supplementary Fig. 1a). The notion that the expression of the GAr promotes cell proliferation was supported by fluorescence-activated cell sorting (FACS) analysis showing an average 30% increase of cells in S-phase expressing the GAr under normal growth conditions. The GAr-dependent increase in proliferation was further enhanced (from 11 to 19%) in cells that had reached confluence and the most notable increase was observed under conditions of low serum when the expression of the GAr resulted in a near threefold increase from an average 3.6 to 11.1% (Fig. 1b and Supplementary Fig. 1b). The GAr has a more prominent effect on translation inhibition when fused to the 5′, as compared to the 3′, end of open reading frames (ORFs) (Fig. 1c, lower panels). Expression of three different constructs carrying the GAr in the N terminus (the chicken ovalbumin (Ova), p53 or GFP) induced an increase of cells in S-phase ranging from 18, 27 and 40%, respectively. However, GAr fused to the C terminus had little effect on cell proliferation, despite the GAr-fusion protein being expressed at high levels (Fig. 1c, upper graph). Similarly, when altered GAr sequences were expressed in which the translation disrupting capacity was compromised by changes in the repeat sequence[28], we observed a direct correlation between cell proliferation and translation inhibition (Supplementary Fig. 1c). Thus, the presence of the GAr in the cells per se does not have an effect on cell proliferation and only when it attenuates translation. Taken together, these results show a link between GAr-mediated suppression of mRNA translation in cis and cell proliferation.

The link between EBNA1 and c-Myc is manifested in Burkitt's lymphoma by the translocation of *c-myc* gene to the immunoglobulin (Ig) locus but the correlation between c-Myc and EBNA1 under normal conditions has remained an enigma. When we expressed an increasing amount of full-length GAr (0.2–1.0 μg DNA in transfection) in H1299 cells, we observed a dose-dependent increase in c-Myc levels (Fig. 1d). We observed a similar increase in c-Myc following expression of GAr in Saos-2 and A549 cells (Supplementary Fig. 2). The *c-myc* 5′ UTR supports cap-independent translation and fusion of this RNA sequence to the 5′ UTR of GAr-Ova (IRES GAr-Ova) is sufficient to overcome translation inhibition without altering the coding sequence, resulting in high levels of GAr-Ova but without the induction in c-Myc expression (Fig. 1e)[19,29]. Expression of a GAr cDNA construct resulted in an approximately sixfold increase in *c-myc* mRNA levels at 16 h following transfection that reached a peak of about tenfold at 36 h. In parallel, the mRNA levels of the c-Myc target gene *CAD* started to increase at 24 h and peaked at

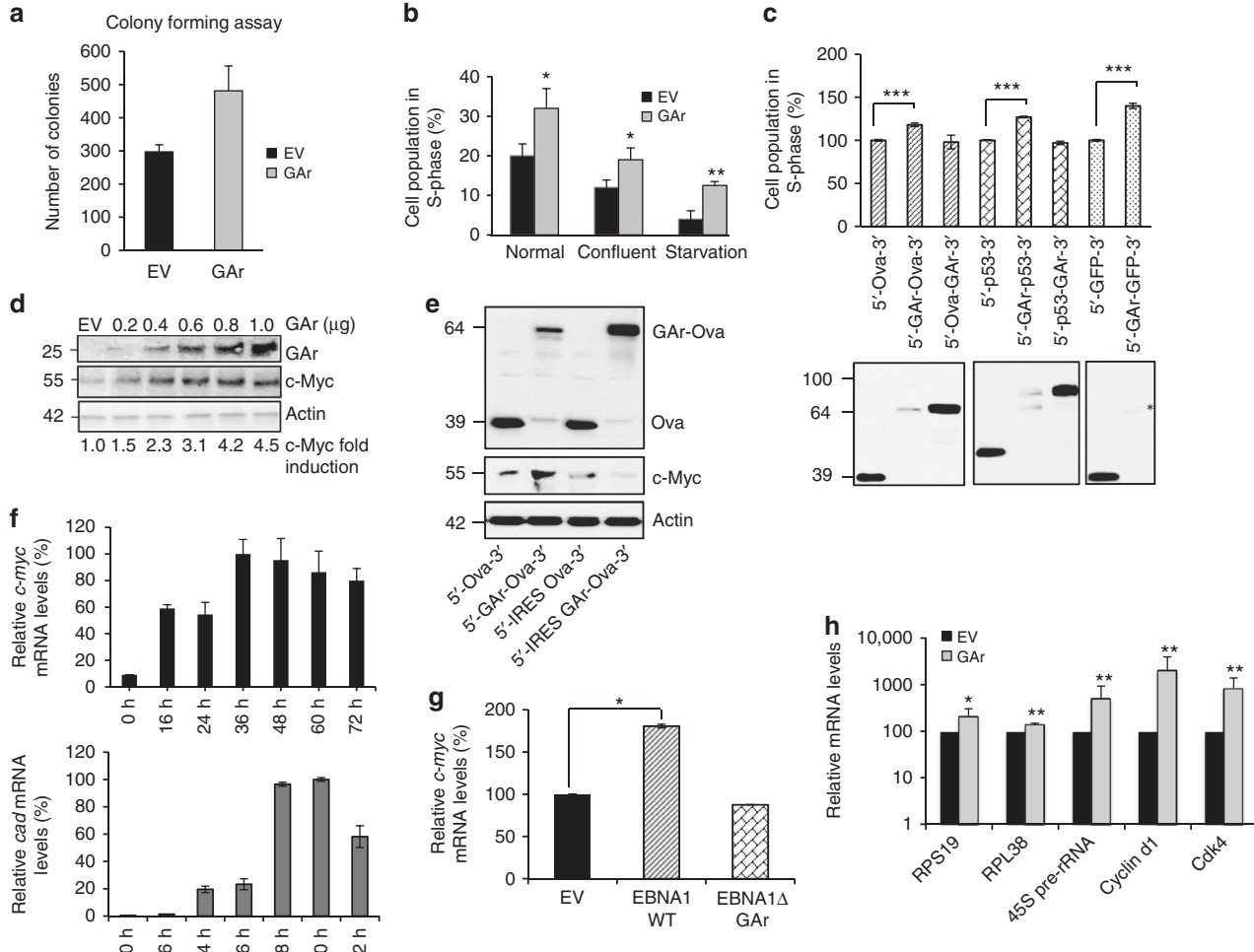

**Fig. 1** GAr-induced mRNA translation stress stimulates cell proliferation via c-Myc. **a** Colony forming assay. The number of colonies of H1299 cells expressing the gly-ala repeat (GAr) of the EBV-encoded EBNA1, or plasmid control (EV), were counted after 15 days in selection medium and plotted. **b** The graph shows the percentage of proliferating cells (S-phase) expressing the GAr or EV under indicated growth conditions as determined by FACS analysis. **c** Western blots (lower panel) show that fusing the GAr to the N terminus of the open reading frames of chicken ovalbumin (Ova), (GAr-Ova), p53 (GAr-p53) or GFP (GAr-GFP) suppresses mRNA translation, in contrast to when inserted at the C-terminal end (Ova-GAr or p53-GAr). The graph (upper panel) shows the relative number of cells in S-phase expressing the indicated constructs. **d** Western blots show the expression of c-Myc following expression of the indicated quantities of GAr cDNA in H1299 cells. The numbers below the blots indicate c-Myc levels normalised to actin. **e** Fusing the 5′ UTR of *c-myc* that contains IRES to the 5′ of GAr-Ova (IRES GAr-Ova) abrogates translation suppression and averts c-Myc induction. **f** The upper graph shows RT-qPCR of *c-myc* mRNA levels at indicated time points following expression of the GAr. The data show the relative *c-myc/GAPDH* mRNAs ratio and the highest value at 36 h is set to 100%. The lower graph shows the relative values of c-Myc-induced *CAD* mRNA levels relative to *GAPDH*. The highest value is set to 100%. **g** RT-qPCR analysis show the relative *c-myc* mRNA levels normalised to GAPDH in H1299 cells expressing EV, EBNA1 WT or EBNA1 ΔGAr. **h** RT-qPCR analysis of *c-myc* target genes (Cdk4, cyclin d1, 45S pre-rRNA, Rps19 and Rpl38) in cells expressing EV or GAr. Note the logarithmic scale. FACS analyses show mean values with s.d. from at least three independent experiments. Actin was used as a loading control. For western blots, RT-qPCR and colony formation assays representative of three independent experiments were shown with s.d. Statistical significance was calculated using *t* tests (***$p < 0.001$, **$p < 0.05$ and *$p < 0.1$)

48 h (Fig. 1f, upper and lower graphs). Similarly, when we expressed EBNA1 WT in H1299 cells, there was a significant induction of c-Myc, whereas an EBNA1 construct lacking the GAr sequence (EBNA1 ΔGAr) had no effect, confirming the importance of the GAr in c-Myc induction (Fig. 1g). We also observed a significant increase in mRNA levels of c-Myc Pol II (*Cdk4, Cyclin D1, Rps19* and *Rpl38*) and Pol I (*45S pre-rRNA*) target genes following GAr expression (Fig. 1h). When the c-Myc inhibitor F4-10058 was added, we observed an ~80% suppression of cells in S-phase at 16 μM in control cells, whereas the corresponding suppression in GAr-expressing cells at the same concentration was about 40% (Supplementary Fig. 3a). However, the suppression profile of *CAD* mRNA levels by F4-10058 was similar in cells expressing the GAr compared to control cells

(Supplementary Fig. 3b). This discrepancy between preventing cell proliferation and c-Myc target genes indicates that other GAr-induced factors are promoting cell proliferation. The treatment with F4-10058 did not prevent the GAr from inducing c-Myc expression (Supplementary Fig. 3c).

**Induction of *c-myc* mRNA levels by GAr is mediated by E2F1.** In order to identify the GAr-dependent factor activating the *c-myc* promoter, a series of constructs that include the P1 and P2 of the *c-myc* promoter were fused to a *Renilla luciferase* reporter gene. Three constructs spanning from (−2052 to +34), (−462 to +34) and (−107 to +34), all induced *luciferase* mRNA levels following expression of the full-length GAr (Fig. 2a). The induction

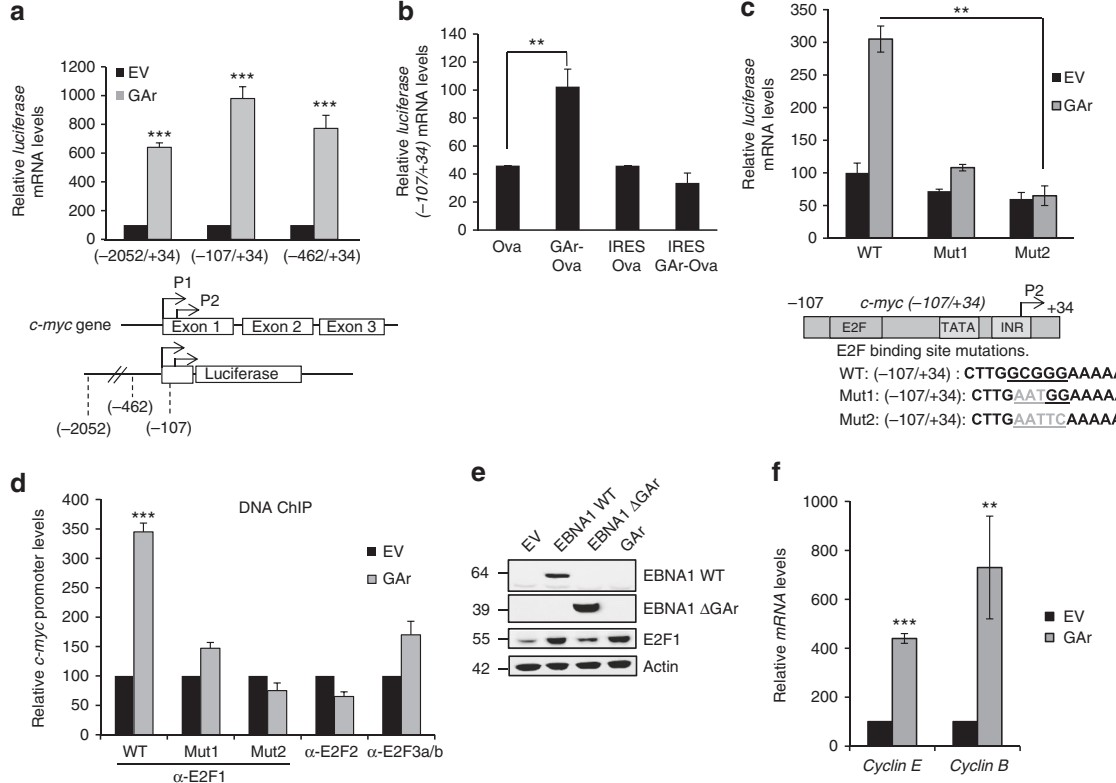

**Fig. 2** E2F1 promotes GAr-dependent induction of *c-myc*. **a** RT-qPCR data show the relative induction of *c-myc-luciferase (Renilla)* reporter constructs (indicated in lower panel) following expression of the GAr, as compared to control plasmid (EV). **b** Relative RT-qPCR values show the induction of *c-myc* promoter activity from the (−107/+34) luciferase reporter construct using indicated Ova and GAr-Ova (±IRES) cDNA constructs. **c** Relative RT-qPCR values from the (−107/+34) luciferase reporter constructs carrying either three (Mut1) or five (Mut2) point mutations in the E2F1 consensus binding site (indicated in lower panel). **d** DNA chromatin IP (ChIP) assay using the (−107/+34) sequence of the *c-myc* promoter constructs (WT, Mut1 and Mut2) and antibodies against E2F1, E2F2 or E2F3 from cells expressing the GAr or EV. **e** Western blots showing the E2F1 levels in cells expressing EV, EBNA1 WT, EBNA1 ΔGAr or GAr with actin used as loading control. **f** RT-qPCR data of the induction of the E2F1 target genes *Cyclins E* and *B* following expression of GAr. For RT-qPCR, the values were normalised with GAPDH. The values represent the mean data from three independent experiments with s.d. Significance was calculated using *t* tests (***$p < 0.001$ and **$p < 0.05$)

of the *Renilla* reporter gene (−107/+34) did not take place when the translation inhibitory capacity of the GAr was abolished by placing the *c-myc* IRES in the 5′ UTR (Fig. 2b). All constructs include the E2F-binding site and the capacity of the GAr to activate the shortest *c-myc* promoter construct (−107 to +34) were severely hampered when three nucleotides were mutated (Mut1) in the E2F-binding site and completely abolished when five mutations were introduced (Mut2) (Fig. 2c). To further address the role of E2F in the GAr-mediated induction of c-Myc, we carried out DNA ChIP assays on the c-myc (−107 to +34) reporter constructs and we used antibodies against E2F1, E2F2 or E2F3. Quantitative PCR revealed an over threefold increase in E2F1 binding to the *c-myc* promoter following expression of the GAr and minor binding with E2F2 and E2F3. This was precluded when we use E2F-binding site mutant (Mut1 and Mut2) constructs (Fig. 2d). We also carried out ChIP assays on the endogenous *c-myc* promoter and we confirmed an increase in E2F1 binding following GAr expression (Supplementary Fig. 4a). The GAr-mediated activation of the *c-myc* promoter was also observed in mouse embryo fibroblast (MEF) cells (Supplementary Fig. 4b). Also, when we silenced E2F1 using siRNA, we observed a decrease in c-Myc levels both in normal and GAr-expressing cells (Supplementary Figs. 4c and 4d). Furthermore, expressing EBNA1 WT, as compared to EBNA1 ΔGAr, resulted in a significant increase in E2F1 expression (Fig. 2e and Supplementary Fig. 4e, f). Next, we tested the GAr effect on other E2F1

transcriptional target genes and could observe an approximately four- and sevenfold increase in the levels of *Cyclin E* and *Cyclin B* mRNAs (Fig. 2f).

**GAr triggers post-transcriptional increase of E2F1 levels.** We next set out to understand how E2F1 is activated by GAr-mediated translation stress. Quantitative RT-PCR following expression of GAr resulted in an over sixfold induction of *c-myc* mRNA levels, without affecting the levels of *E2F1* mRNA, suggesting that activation of E2F1 is post-transcriptional (Fig. 3a).

Viral factors such as HPV E7, adenovirus E1A and SV40 large T activate E2F by preventing the interaction with the pRb protein. As the expression levels per se are not important for GAr-mediated induction of E2F1 it is unlikely that the GAr would mimic the function of these viral proteins and prevent the E2F1–pRb interaction. In line with this, we observed that overexpression of pRb resulted in a complete suppression of GAr-mediated activation of E2F1, as measured by *Cyclin E* mRNA levels. However, whereas 0.2 μg of pRb cDNA was sufficient to inhibit E2F activity in the control cells, a threefold excess was required to achieve a similar effect in cells expressing the GAr (Fig. 3b). The hypothesis that the GAr is not targeting the pRB–E2F interaction was further supported by the observation that overexpression of HPV E7 resulted in a strong induction of E2F activity in control cells but resulted in only a marginal further increase in E2F activity in cells expressing the GAr-Ova

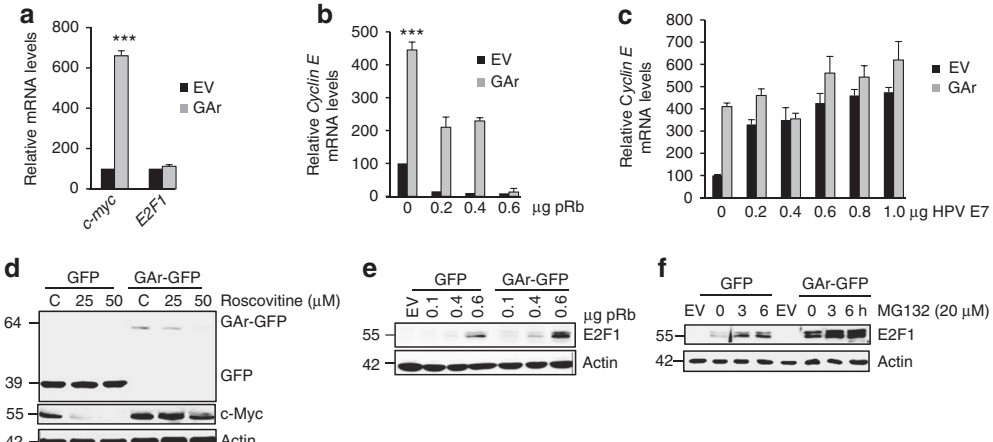

**Fig. 3** GAr induces E2F1 protein levels. **a** RT-qPCR data show the relative levels of *c-myc* and *E2F1* mRNA following GAr expression. **b** The effect of GAr-induced E2F1 activity as determined by RT-qPCR on *Cyclin E* mRNA levels following increasing amounts of pRb. **c** As in **b** but with increasing amounts of the pRb binding HPV16-E7. **d** The effect on GAr-mediated induction of c-Myc expression following treatment with increasing concentrations of the cell cycle kinase inhibitor roscovitine. **e** Western blots showing the levels of E2F1 in cells expressing GFP or GAr-GFP following increasing amounts of pRb. **f** The levels of E2F1 expression following treatment with the proteasome inhibitor (MG132). The data presented in **a**, **b** and **c** show the means from three independent experiments with s.d. The mRNA levels were normalised against GAPDH and the reference value 100 was set for non-transfected cells. Statistical significance was calculated using *t* tests (***$p < 0.001$). Western blots represent $n \geq 3$ and actin was used as a loading control

(Fig. 3c). During the transition from $G_0$ to $G_1$, pRb becomes phosphorylated by cyclin-dependent kinase (CDK) activity, which results in the release of E2F1. When we treated control cells with the CDK inhibitor roscovitine, we observed reduced levels of c-Myc but this effect was significantly less in cells expressing the GAr (Fig. 3d). Free active E2F1 has a higher turnover rate and these three observations are consistent with higher levels of an active E2F1 in GAr-expressing cells. In support of this, we observed that an increase in the levels of pRb resulted in an increase in E2F1 protein levels in cells expressing the GAr-GFP compared to cells expressing GFP alone (Fig. 3e). A similar increase in E2F1 levels was observed using the proteasome inhibitor (MG132) (Fig. 3f). We did not observe E2F1 induction when GAr was expressed under c-myc IRES control (Supplementary Fig. 4g).

**Induction of E2F1 is mediated via its coding sequence.** An increase in E2F1 protein synthesis by the GAr was observed using ${}^{35}$S-metabolic pulse labelling followed by E2F1 immunoprecipitation (Fig. 4a). As the mRNA levels of *E2F1* do not change following expression of the GAr, this is consistent with the GAr triggering an increase in the rate of E2F1 mRNA translation. Expression of E2F1 constructs that include the 5′ and the 3′ UTRs (E2F1 UTR), or the coding sequence alone (CDS), resulted in a similar GAr-dependent increase in E2F1 expression, indicating that the increase in the rate of *E2F1* mRNA translation is not mediated by the classic regulation of translation that acts via the untranslated sequences (Fig. 4b). Polysomal profiling followed by RT-qPCR analysis of *E2F1* mRNA showed significant enrichment of *E2F1* mRNA in the polysomal fractions in cells expressing the GAr compared to empty vector (EV), indicative of more ribosomes being linked to the E2F1 mRNA and, thus, a higher rate of protein synthesis (Fig. 4c). Other mRNAs, such as *BiP* and *p21*, did not show an increase in polysome fractions (Supplementary Fig. 5) indicating that the increase in E2F1 level under GAr induction is not a result of global translation increase. Furthermore, deletion mutants of the *E2F1* CDS truncated from the 5′ showed that the first 432 nucleotides of the CDS contain the GAr-responsive element (Fig. 4d). Interestingly, Δ(+1 to +324) showed low expression in the absence of GAr and a strong induction,

suggesting that regulation of E2F1 synthesis is mediated via two RNA domains and might involve the release of a negative acting factor. In support of this, we observed that the deletions Δ(+1 to +525) and Δ(+1 to +585) resulted in higher levels of expression but reduced E2F1 induction (Fig. 4e). Together, these results point towards mRNA structure motif/s being the target for GAr-dependent induction of *E2F1* mRNA translation.

**mRNA translation stress induces E2F1 via PI3Kδ.** The induction of E2F1 synthesis following disruption of mRNA translation on GAr-carrying polysomes is likely to be relayed by a signalling pathway. Some of the described signalling pathways known to affect global cap-dependent mRNA translation act via mTOR and the phosphorylation of 4E-BP1, or via S6 kinase[30]. However, 8 h treatment with the mTOR inhibitor rapamycin, or with AKT-inhibitors, had no effect on GAr-mediated induction of E2F1 expression (Fig. 5a and Supplementary Fig. 6). This is in line with data showing that the trans-effect of the GAr on mRNA translation is mediated via the CDS and not via sequences in the UTRs. We also tested inhibitors against the MAPK, cell cycle kinases, DNA-PK, ATM and several other kinases linked to growth control but with little effect. However, when we tested two general PI3K inhibitors, wortmannin and LY94002, we observed a complete inhibition of GAr-mediated induction of E2F1 (Fig. 5a and Supplementary Fig. 7a). Inhibitors of different PI3Ks catalytic subunits showed that the PI3Kδ-specific inhibitor CAL-101, but not inhibitors of p110α, p110β or p110γ, suppressed GAr-mediated induction of E2F1 (Fig. 5b and Supplementary Figs. 6 and 7a). Wortmannin efficiently prevented phosphorylation of PI3K downstream targets AKT and the mTOR substrate 4E-BP1 in cells, where PI3K activity had been induced following treatment with 40 ng/ml of PDGF for 30 min in serum-free medium. However, CAL-101 had less effect on phosphorylation of AKT and little effect on phosphorylation of 4E-BP1 under these conditions (Fig. 5c). In line with this, there was no GAr-dependent effect on the phosphorylation of AKT and 4E-BP1, showing the specificity of PI3Kδ activation (Fig. 5d and Supplementary Fig. 6). A role for p110δ in the control of E2F1 expression was further supported using siRNA against p110δ. Importantly, suppression of p110δ expression using siRNA also reduced E2F1 levels in

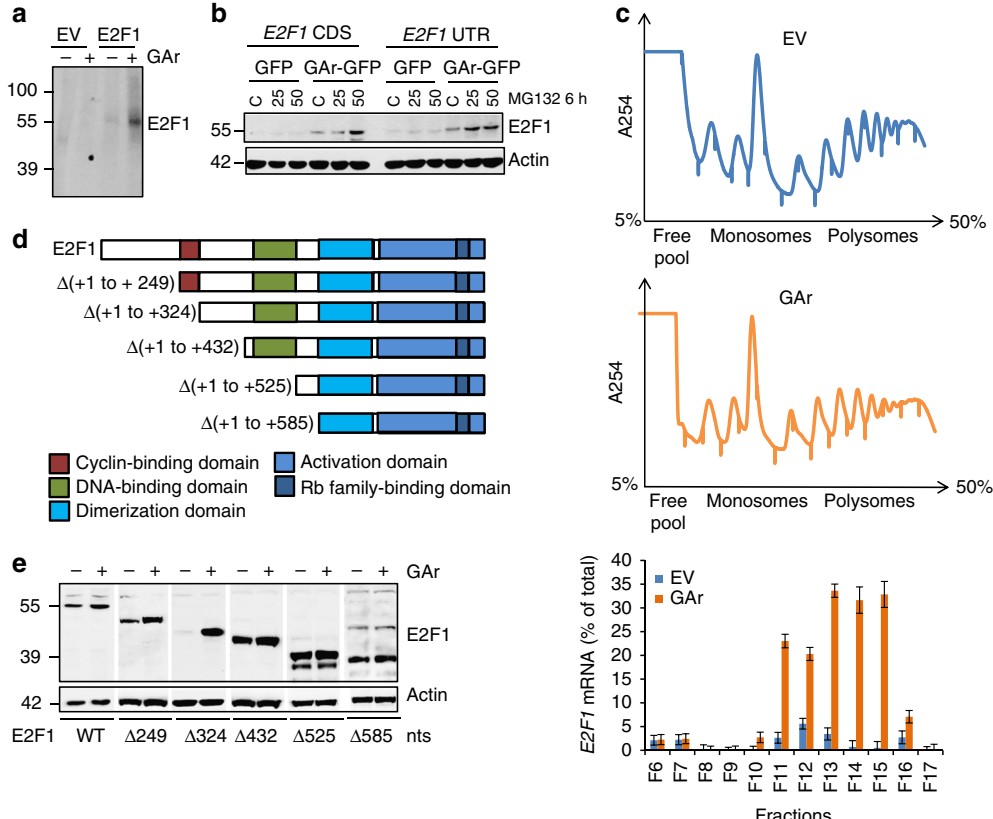

**Fig. 4** GAr-induced mRNA translation stress stimulates E2F1 synthesis. **a** Autoradiograph shows the effect of GAr on newly synthesised E2F1 levels following [35]S-Met pulse labelling in the presence of a proteasome inhibitor (20 μM MG132) followed by E2F1 immunoprecipitation. **b** Western blots showing GAr-dependent induction of E2F1 from constructs carrying the untranslated (UTR) regions of the *E2F1* mRNA (E2F1 UTR), or the coding sequence alone (E2F1 CDS). **c** Polysome profiling and RT-qPCR analysis of *E2F1* mRNA in cells expressing the GAr or EV. The upper panels show polysome profiles. The lower panel shows the relative E2F1 mRNA levels (%) from total mRNA normalised against actin. The values represent the mean data from three independent experiments with s.d. **d** Schematic representation of functional domains of E2F1 deleted from the 5′ of the CDS. The +1 indicated the first AUG of the CDS. **e** Western blots showing the expression of E2F1 deletion mutants with and without GAr expression. Western blots represent n ≥ 3, actin was used as a loading control

non-GAr-treated cells, indicating that the p110δ-dependent E2F1 regulatory pathway is not unique to EBNA1 activity (Fig. 5e and Supplementary Fig. 7b). GAr-dependent induction of E2F1 is further stimulated following treatment with general PI3K stimulant (LPS [1 μg/ml] and MgSO$_4$ [20 mM])[31] and was completely suppressed by the addition of CAL-101 (10 μM). This was also observed in EBV-infected lymphoblastoid B95-8 cells (Fig. 5f and Supplementary Figs. 6 and 7c). In line with this, E2F1 levels were reduced significantly upon CAL-101 treatment in H1299 cells expressing EBNA1 WT as compared to EBNA1 ΔGAr (Supplementary Fig. 7d). There are no apparent changes in the expression levels and localisation pattern of p110δ in cells expressing the GAr (Fig. 5d and Supplementary Fig. 8).

**PI3Kδ mediates EBNA1 oncogenic activity.** Several oncogenic viruses target the E2F1 pathway but this has not yet been shown for the EBV. Nevertheless, EBNA1 has been implicated in having an oncogenic activity and a transgenic EBNA1 animal model in which EBNA1 transgene expression was driven by the immunoglobulin heavy chain (EμEBNA1) showed an inverse correlation between EBNA1 expression and lymphoma phenotype[14]. To test if the oncogenic properties of EBNA1 are related to activation of p110δ, we used an EμEBNA1 mouse-derived primary lymphoma cell line. Treatment with CAL-101 in culture showed a dose-dependent increase in cell death starting at 10 μM (Fig. 6a). Three further independent primary tumours from

different mice of the same transgenic line (EμEBNA1.26)[14] showed a CAL-101-mediated cell death starting at 2 days of treatment (10 μM) (Fig. 6b). EBNA1 transgenic cells showed increased expression of both c-Myc and E2F1 compared to the non-transgenic wild-type (WT) lymphocytes (Fig. 6c and Supplementary Figs. 9a and 9b) and we observed loss of both E2F1 and c-Myc expression after 1 day of treatment at 10 μM (Figs. 6c, d and Supplementary Fig. 9c). To test if CAL-101-mediated cell death is linked to EBNA1 expression, we compared a cell line derived from an LMP1 and EBNA1 bi-transgenic lymphoma (3959.48) to an LMP1-only cell line (39.415)[14,32]. The LMP1 and EBNA1 bi-transgenic cells are EBNA1-dependent and treatment with CAL-101 at 50 μM resulted in death, while the EBNA1-independent (39.415) cell line showed no increase in cell death (Fig. 6e). We also treated the EBV-positive Burkitt's lymphoma (BL) Raji cells and EBV-negative BL41 cells with CAL-101 and we observed a suppression of E2F1 in Raji but with less effect in BL41 (Fig. 6f). Raji cells carry the BL characteristic *c-myc* gene Ig translocation, whereas B95.8 is an EBV-transformed lymphoblastoid cell line and do not have the *c-myc* translocation. Only in the B95.8, and not in the Raji cells, did we observe a CAL-101-dependent suppression of c-Myc expression (Fig. 6g). These results support the hypothesis that EBNA1 exerts an oncogenic activity via GAr-dependent activation of E2F1 and the consequent induction of c-Myc.

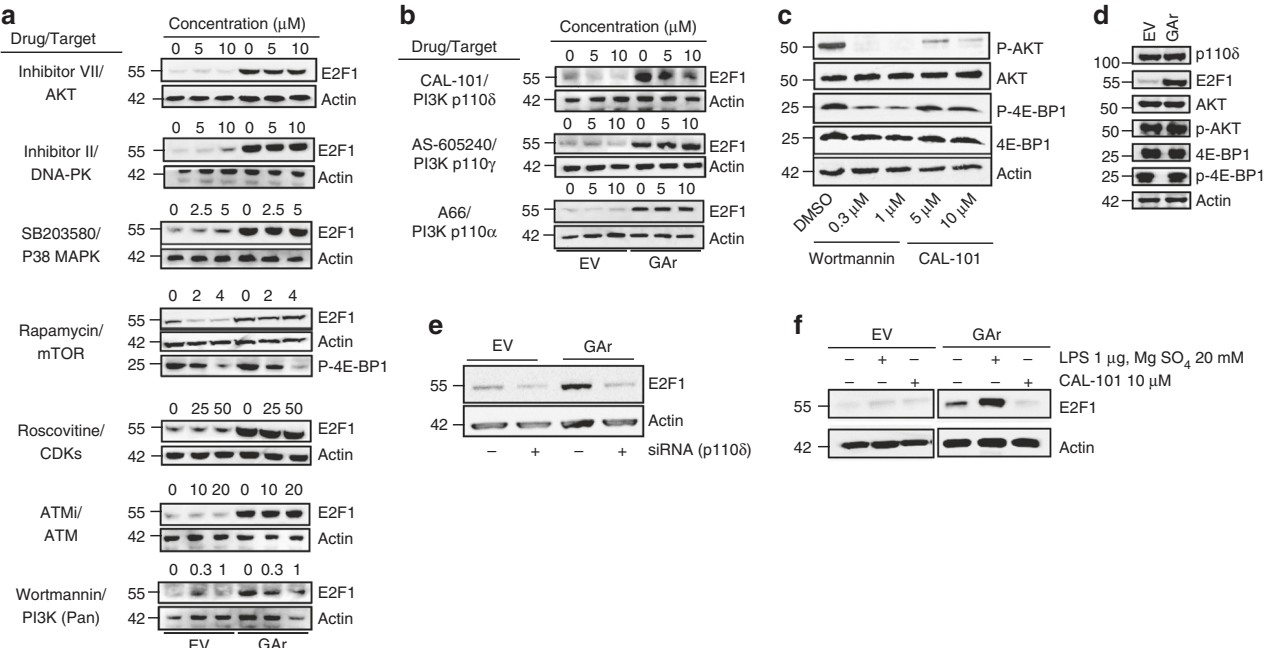

**Fig. 5** GAr-dependent induction of E2F1 is mediated via PI3Kδ. **a** Western blots show the expression of E2F1 in H1299 cells expressing the GAr or EV following treatment with indicated drugs for 8 h. Phosphorylated 4E-BP1 (P-4E-BP1) was used to show mTOR activity. **b** Western blots show E2F1 expression levels in cells expressing the GAr or control plasmid (EV), treated with the indicated concentrations of inhibitors of PI3K p110δ (CAL-101), p110γ (middle panel) and p110α (lower panel) subunits. **c** The effect of wortmannin and the PI3Kδ-specific inhibitor (CAL-101) on the phosphorylation of AKT (P-AKT) and 4E-BP1 (P-4E-BP1) in cells treated with 40 ng/ml of PDGF for 30 min in serum-free medium. **d** Western blots showing the effect of GAr on the phosphorylation of AKT and 4E-BP1. **e** Western blot showing the expression of E2F1 following 24 h treatment with siRNA against PI3Kδ. **f** Western blot showing the expression of E2F1 in cells expressing EV or GAr followed by general induction of PI3K activity using $MgSO_4$ (20 mM) and LPS (1 μg), with, or without, CAL-101 (10 μM). Western blots represent $n \geq 3$. Actin was used as a loading control.

## Discussion

By disrupting mRNA translation in cis using the unique properties of the gly-ala repeat of the EBNA1 we can show that cells respond to mRNA translation stress by a PI3Kδ-dependent activation of E2F1 synthesis and the induction of c-Myc. We have also shown how this pathway accounts for EBNA1 oncogenic activity in a transgenic animal model. It is well described how upregulation of c-Myc induces ribosomal biogenesis to promote protein synthesis and these results illustrate a feedback pathway from the ribosome to c-Myc, whereby the cell senses the status of the mRNA translation and triggers a pathway aimed at restoring dysfunctional protein synthesis. We propose that this pathway is exploited by the EBV to promote cell growth (Fig. 7).

The pool of induced E2F1 is active and not bound to pRb. In line with this, we found that overexpression of the papilloma E7 protein had only a marginal effect on E2F1 activity in GAr-expressing cells. The induction of E2F1 by PI3Kδ is at the level of increased mRNA translation and is unusual in the sense that it is mediated via the coding sequence of the *E2F1* mRNA, and not via the more commonly used 3′ and 5′ UTRs. This indicates an mRNA-specific regulation that helps to explain how broad inhibitors of the PI3K catalytic subunits, such as wortmannin and LY294002, can efficiently block GAr-dependent induction of E2F1, while inhibitors of AKT or mTOR that control general cap-dependent translation have little, or no effect. We are currently trying to identify the PI3Kδ substrate that controls E2F1 synthesis and based on the deletion series of the *E2F1* coding sequence it is likely this factor is an RNA structure-binding protein. This shows conceptual similarity to the induction of p53 synthesis following DNA damage, which is also mediated by a stress-responsive RNA structure in the coding sequence of the *p53* mRNA[33]. The role of PI3Kδ in the mRNA translation stress response was unexpected

as this family of kinases is normally associated with extracellular growth stimulatory factors and activation by tyrosine kinase or G-protein-coupled receptors. However, a hint that PI3Kδ might play a role in the EBV life cycle is suggested by the observation that patients who develop the gain-of-function condition, activated PI3Kδ syndrome (APDS), typically suffer from chronic EBV (and other herpesviruses) viraemia[34]. It is interesting that suppression of PI3Kδ inhibits E2F1 expression also in cells that do not express the GAr, which implies a broader role of this pathway in controlling E2F1 expression. It is, thus, likely that cellular messages also trigger this pathway and that a high rate of protein synthesis can help explain PI3Kδ activity also in solid cancers. PI3Kδ inhibitors have been effective in treating chronic lymphocytic leukaemia (CLL) and indolent Hodgkin's lymphoma and these results help to better understand the underlying molecular mechanisms. Hodgkin's lymphomas are associated with EBV in ~40% of cases and it will be interesting to see if there is a correlation between EBV-carrying cancers and Idelalisib efficacy. In support of this, we observed that bi-transgenic lymphoma cells depending on EBNA1 are sensitive to CAL-101 treatment. Nasopharyngeal carcinomas that carry the EBV and express EBNA1 in 100% of the cases is another type of cancer in which this pathway might be of therapeutic interest.

Activation of E2F1 is a common strategy by oncogenic viruses to promote cell proliferation but this has previously not been demonstrated for EBV. Papilloma, adenovirus and simian SV40 viruses have developed strategies to activate E2Fs by expressing factors that disrupt the interaction with pocket proteins. The EBNA1 has evolved a different route that instead exploits a cellular pathway that aims to restore ribosomal activity via activation of E2F1 under conditions of dysfunctional ribosomal activity. The same inhibition of mRNA translation in cis by the GAr forms

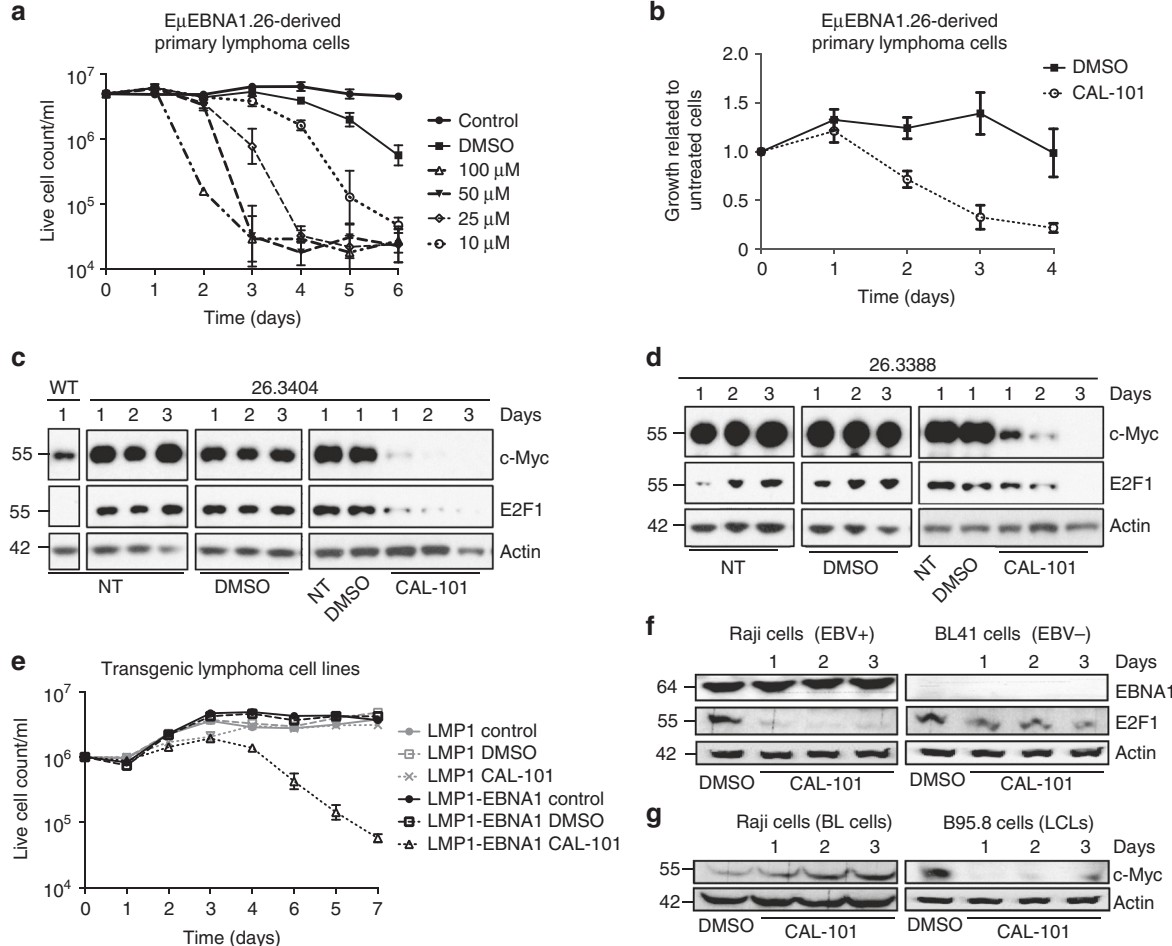

**Fig. 6** E2F1, c-Myc expression and cell viability are mediated via PI3Kδ in EBNA1 transgenic mouse tumour cells. **a** Primary lymphoma cells derived from mice were cultured (in triplicate) with 10–100 μM CAL-101 or DMSO, or no additive (control) for 6 days. Rapid cell death correlates with increasing concentrations of CAL-101. **b** Cells from three different explanted primary EBNA1-positive transgenic lymphomas were each cultured in triplicate for 4 days with 10 μM CAL-101 added (in 0.01% DMSO) or 0.01% DMSO. The ratio of CAL-101 and DMSO-treated cell counts against untreated cell counts is graphed. **c**, **d** Western blots showing a time course of the expression of E2F1 and c-Myc in non-transgenic cells (WT) and using transgenic cells from two different primary tumours (ID:26.3404 [C] and 26.3388 [D]) (see also Supplementary Fig. 9a). Explanted EμEBNA1 transgenic tumour cells were cultured for up to 4 days, with no treatment (NT), or with vehicle (DMSO) or with daily treatment of 10 μM CAL-101, as indicated. E2F1 and c-Myc expression in the EBNA1 transgenic lymphoma cells are markedly reduced by CAL-101 treatment. **e** Cell growth was measured for 7 days in two mouse B cell lymphoma cell lines LMP1 positive (ID: 39.415) and the bi-transgenic LMP1 and EBNA1 positive (ID: 3959.48). Cells were cultured with daily addition of CAL-101 to 50 μM, or with vehicle (DMSO) and viable cells counted. **f** Western blots showing the expression of E2F1 in two Burkitt's lymphoma cell lines carrying c-myc gene translocation. Raji is EBV positive and BL41 is EBV negative. Both were treated with CAL-101 (10 μM) for indicated time. **g** Western blots showing c-Myc expression following treatment with CAL-101 (10 μM) in Raji and the lymphoblastoid cell line B95.8 (no c-myc gene translocation). Western blots represent n ≥ 3; Actin was used as a loading control. The values in **a**, **b** and **e** represent the mean data from three independent experiments with s.d

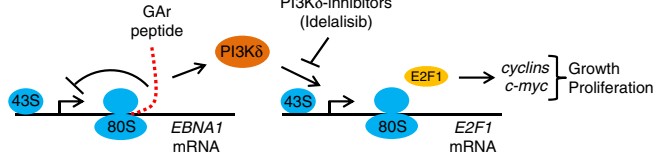

**Fig. 7** mRNA translation stress on EBNA1 polysomes activates PI3Kδ and E2F1 synthesis. Translation attenuation in cis caused by the GAr sequence of the EBNA1 mRNA causes translation stress, which results in activation of PI3Kδ and the induction of E2F1 mRNA translation independent of AKT and mTOR pathways. The newly synthesised E2F1 induces downstream target genes, including cyclins and c-myc, resulting in an increase in cell growth and proliferation

part of the viral strategy to evade the MHC class I-restricted antigen presentation pathway. Hence, targeting mRNA translation allows the virus to hit two birds with one stone: evade the immune system and at the same time support the proliferation of infected cells. Despite being the only EBV antigen expressed in all BL and also consistently expressed in all other EBV-associated tumours such as nasopharyngeal carcinomas, the oncogenic activity of EBNA1 has remained an enigma. An inverse correlation between EBNA1 levels and the penetrance of B cell lymphomas was reported[14,15]. Additionally, c-Myc was found to be a noted player in these EBNA1 tumours, showing upregulated expression (Supplementary Fig. 9a). Importantly, we can now show that E2F1 and c-Myc expression in primary lymphomas derived from the lymphoma-prone EμEBNA1 mice are indeed inhibited following CAL-101 treatment. It is, thus, likely that the reported inverse correlation between the expression levels of EBNA1 and the lymphoma phenotype in the EμEBNA1 animals

can be explained by the induction of E2F1 by EBNA1-mediated mRNA translation stress. In line with this, c-Myc levels decrease after CAL-101 treatment in EBV-infected lymphoblastoid cells, but not in Burkitt's lymphoma (BL) cells where the transcription of *c-myc* is deregulated through translocation to an immunoglobulin locus.

## Methods

**Cell culture and treatments**. Unless mentioned otherwise, experiments were performed mostly using H1299 cells (non-small-cell lung carcinoma human cell line) [NCI-H1299 (ATCC® CRL5803™)]. Other cell lines used were Raji (type III latency Burkitt's lymphoma), B95.8 lymphoblastoid cell line. All cell lines were cultured in RPMI 1640 medium supplemented with 10% foetal bovine serum, 2 mM L-glutamine, 100 U/ml penicillin, 100 μg/ml streptomycin (Invitrogen) and 5 μg/ml Plasmocin prophylactic (Invivogen). Cell lines were routinely checked for mycoplasma contamination using PlasmoTest kit (Invivogen). Two mouse B cell lymphoma cell lines were used: 39.415 established from an LMP1 driven transgenic mouse tumour and 3959.48 established from an LMP1 and EBNA1-positive bi-transgenic mouse tumour[32]. Primary tumour cells from lymphomas derived from the EμEBNA1 transgenic mouse line 26[14], or in vivo passages of these tumours, or wild-type (non-transgenic) splenocytes were explanted, red blood cells lysed and cultured directly, or frozen for subsequent culture. Starvation experiment allowed the cells to grow for 72 h with complete medium supplemented with 0.1% foetal bovine serum. H1299 stable cell lines were generated through transient transfection of neomycin gene-containing pcDNA3 vectors. After 2 to 3 weeks, geneticin resistant populations were selected and gene stability was verified for at least ten passages.

Lymphoma cells were cultured in RPMI (with L-glutamine, 10% FBS) with 50 μM 2ME in primary cultures. CAL-101 was dissolved in DMSO and supplemented in cultures from 10 to 100 μM daily. Live cell counts and cell viability were determined by trypan blue exclusion and counted using the Countess Automated Cell Counter (Thermo Fisher Scientific).

**Plasmid DNAs and siRNAs**. The full-length GAr cDNA expression vector was isolated from the Epstein–Barr virus (strain B95-8)-encoded EBNA1 and cloned next to ovalbumin cDNA or p53 cDNA. The generated pcDNA3 constructs, GAr, Ova, GAr-Ova, Ova-GAr, p53, GAr-p53, p53-GAr, *c-myc* IRES-Ova and *c-myc* IRES-Ova-GAr, were described in earlier reports[17,19]. The GAr cDNA was cloned in the peGFP-N1 vector generating a GAr-GFP fusion protein with GAr at the NH2 terminal (GAr-GFP). E2F1 coding sequence was amplified from the cmv-E2F1 (provided by Dr O. Bischof, Inst. Pasteur, Paris) and cloned in to pcDNA3 vector. E2F1 CDS deletion constructs were created using pcDNA3-E2F1 (CDS) amplified with corresponding primers (listed in Supplementary Table. 1) and cloned into pcDNA3 vector. The pRB construct was provided by Dr O. Bischof and CMV-HPV16-E7 is an expression vector for high-risk HPV16-E7 oncoprotein that was provided by Dr K. Münger (Brigham & Women's Hospital, MA). The pGL2-Basic luciferase reporter vector containing the human *c-Myc* promoter (−2052/+34), (−462/+34) and (−107/+34) were kindly provided by Professor LZ Penn (Toronto Medical Discovery Tower, ON, Canada).

Using site-directed mutagenesis, sequential deletions of the human *c-myc* promoter (−107/+34) were performed to generate two mutated DNA fragments, designated (−107/+34) Mut1 and (−107/+34) Mut2. For silencing, E2F1 siRNA 1 (2970963, Qiagen), E2F1 siRNA 2(2824210, Qiagen), AllStars scramble siRNA (SI03650318, Qiagen), p110δ siRNA 1 (2989964, Qiagen), p110δ siRNA 2 (2996913, Qiagen), E2F3 siRNA (2965187, Qiagen) were used. For transfection of plasmid DNA and siRNA, GeneJuice (EMD Chemicals) and INTERFERin (Polyplus-transfection) reagents were used according to the manufacturer's instructions.

**Western blotting**. Cells were lysed in BC200 lysis buffer (200 mM NaCl, 0.2% NP-40, 10% (v/v) glycerol, 1.0 mM dithiothreitol (DTT), 1.0 mM EDTA, and 25 mM Tris-HCl, pH 7.8) containing 1% (v/v) eukaryotic protease inhibitor cocktail (Calbiochem). Equal protein amounts were loaded and resolved in 4–12% Bis–Tris Plus Gels (Thermo Fisher), transferred on BioTrace NT pure nitrocellulose blotting membrane (PALL Corporation) and blocked with 5% non-fat dry milk in Tris-buffered saline pH 7.6 containing 0.1% Tween-20. Proteins were then probed with corresponding antibodies; anti-E2F1 rabbit pAbs [1:1000] (C-20, Santa Cruz), anti-GA rabbit pAbs raised against the gly-ala sequence of EBNA1 protein [1:500], anti-p53 rabbit pAbs [CM-1] [1:1000], anti-GFP mouse mAbs [1:1000] (clones 7.1 and 13.1, Roche), anti-chicken egg ovalbumin rabbit pAbs [1:1000] (Sigma-Aldrich), anti-c-Myc mouse mAbs [1:500] (9E10, Santa Cruz and Y69 Abcam: #32072) and anti-actin mouse pAbs [1:2000] (AC-15, Sigma-Aldrich), anti-p110δ rabbit pAb [1:1000] (H219 Santa Cruz), anti-AKT rabbit mAb [1:1000] (2938, Cell Signaling), anti-P-AKT rabbit mAb [1:500] (9271, Cell Signaling), anti-4E-BP1 rabbit mAb [1:1000] (9644, Cell Signaling), anti-p-4E-BP1 rabbit mAb [1:1000] (2855, Cell Signaling). Uncropped blots are available in supplementary information.

**Colony formation assays**. Twenty-four hours post transfection, $1.10^5$ pcDNA3 (vector control) or pcDNA3-GAr–transfected H1299 cells were trypsinized and selected in growth medium supplemented with geneticin. Surviving colonies were allowed to grow for 15 days and stained with 4% (v/v) Giemsa stain. Colonies were then quantified using Clono counter programme.

**Flow cytometry**. Cells expressing indicated constructs were fixed by overnight incubation in cold 70% ethanol. After 30 min incubation with 100 U/ml RNase A (Sigma-Aldrich) at 37 °C, propidium iodide (10 μg/ml)–stained cells were analysed with an LSR flow cytometer and CellQuest software (Becton-Dickinson).

**ChIP assays**. ChIP assays were performed using a kit according to the manufacturer's instructions (Upstate). The cleared samples were incubated overnight with the anti-E2F1 (H137) anti-E2F2 (sc-632) or anti-E2F3 (sc-56666) rabbit polyclonal antibodies. Antibody/protein/DNA complexes were then co-precipitated with the beads according to the kit recommendations and the DNA was recovered by phenol/chloroform extraction and ethanol precipitation. The P2 promoter region (−83/+30) of the human *c-myc* gene was amplified by qPCR (see Supplementary Table. 1 for primer sequences).

**Quantitative reverse transcription-PCR**. Total RNA was extracted with RNeasy Mini Kit (Qiagen) following the manufacturer's instructions. cDNA synthesis was carried out using the Moloney murine leukaemia virus reverse transcriptase and Oligo(dT) Primer (Invitrogen). RT-qPCR was performed on StepOne real-time PCR system (Applied Biosystems) using Perfecta SYBR Green FastMix, ROX (Quanta Biosciences) (See Supplementary Table. 1 for target primer sequences).

**Polysome profiling**. Five–fifty percent wt/vol linear sucrose gradients were freshly casted on SW41 ultracentrifuge tubes (Beckmann) using the Gradient master (BioComp instruments) following the manufacturer's instructions. Forty-eight hours post transfection, H1299 cells (with 80% confluency) were treated with cycloheximide 100 μg/ml for 5 min at 37 °C and then washed twice with 1× PBS (Dulbecco modified PBS, GIBCO) containing cycloheximide 100 μg/ml. Cells were then scrapped and lysed with polysome lysis buffer (100 mM KCL, 50 mM HEPES-KOH, 5 mM MgCl₂, 0.1% NP-40, 1 mM DTT, cycloheximide 100 μg/ml, pH 7.4). Lysates were then loaded on a sucrose gradient and centrifuged at $222228 \times g$ for 2 h at 4 °C in a SW41 rotor. Samples were then fractionated using Foxy R1 fraction collector (Teledyne ISCO) at 0.5 min intervals[35,36]. RNA purifications from fractions were performed using Trizol-LS reagent (Invitrogen) combined with ethanol precipitation. Reverse transcription for the target mRNAs were carried out using equal volume of DNA free RNA from fractions and then normalised with actin levels.

**Metabolic pulse label**. Twenty-four hours post transfection, cells were cultured for 1 h in Dulbecco's modified Eagle's starvation medium (Sigma-Aldrich) (without methionine, cysteine and L-glutamine supplemented with 2% dialysed foetal bovine serum together with 20 μM of proteasome inhibitor MG132. Cells were metabolically labelled for 1 h with 45 μCi/ml of EasyTag Express $^{35}$S-methionine Protein Labelling Mix (Perkin-Elmer). E2F1 protein was then immunoprecipitated and the newly synthesised protein were resolved in 4–12% Bis–Tris Plus Gels and visualised on autoradiograph.

**Immunofluorescence staining**. Cells were grown in cover slips for 24–48 h (for growing non-adherent cells, cover slips were treated first with 0.01% poly-L-lysine solution [Sigma]) and fixed with 4% paraformaldehyde (for H1299 cells were fixed 48 h post transfection), permeabilized with 0.4% Triton X-100, 0.05% CHAPS in PBS. Cells were then blocked using 3% BSA, 0.1% saponine in PBS and incubated with anti-PI3Kδ mouse mAb [1:200] (sc-136032) for 2 h at RT and with ALEXA-488 anti-mouse antibody [1:500] for 45 min at RT, stained with DAPI and washed with PBS. Cover slips were then dried and mounted in slide using DAKO fluorescent mounting medium. Cells were then analysed by Zeiss Axiovert inverted microscope.

**Drugs**. MG132 (474790-5, Calbiochem), ATMi (118500, Calbiochem), DNA-PK Inhibitor II (260961, Calbiochem), AKT Inhibitor VIII (124018, Calbiochem), Rapamycin (553210, Calbiochem), SB203580 (559389, Calbiochem), Wortmannin (681675, Calbiochem), F4-10058 (F3680, Sigma-Aldrich), Roscovitine (R7772, Sigma-Aldrich), AS-605240 (S1410, Selleck Chemicals), CAL-101 (S2226, Selleck Chemicals), A-66 (S2636, Selleck Chemicals), LY294002 (1130, Tocris Bioscience), LPS LipoPolySaccharide (L2880, Sigma-Aldrich).

**Statistical analysis**. Statistical significance was analysed by comparing data with reference point using two-tailed *t* tests, $p < 0.05$ was considered statistically significant (***$p < 0.001$, **$p < 0.05$ and *$p < 0.1$). All statistical assessment was performed using the Microsoft excel programme.

**Data availability**. The authors declare that data supporting the findings of this study are available within the article and supplementary file, or available from the corresponding author on reasonable request.

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

## Acknowledgements

This work was supported by Equipe Labellisée la Ligue Contre le Cancer, the Inserm, the RECAMO projects GACR P206/12/G151, MEYS-NPS I-L01413, Cancerforskningsfonden Norr and Cancerfonden 160598. S.V.G. was supported by la Ligue Contre le Cancer and S.P. by the ARC and PACRI. We thank Dr O. Bischof for providing us cmv-E2F1 and pRb constructs, Dr K. Münger for cmv-HPV16-E7 construct and Prof. Dr L.Z. Penn for pGL2-*c-myc* promoter-luciferase constructs.

## Author contributions

S.V.G. and S.P. designed and carried out most of the experiments. C.D. designed and carried out experiments. K.A. carried out experiments relating to transgenic murine cells and J.B.W. supervised this aspect of the project and contributed to and edited the manuscript. S.V.G. assembled the MS; R.F. supervised the project and assembled the MS together with K.N.

## Additional information

**Competing interests:** The authors declare no competing financial interests.

