## [Peer Review File · Nature Communications]

Reviewers' comments:

Reviewer #1 (Remarks to the Author):

The EBV EBNA1 protein sequence contains a region (GAR) composed of repeated gly and ala amino acids. GAR is not required for the viral DNA replication functions of EBNA1 but is known to play a role in stabilising the EBNA1 protein and in regulating the translation of EBNA1.

There are two main novel features in this paper. The first is the effect of the GAR on PI3Kdelta and the mechanism of how this induces E2F1 activity and c-MYC (Figs 1-5). The second is the growth inhibitory effect of a PI3K delta inhibitor on EBV positive lymphoma cells, linked to the presence of EBNA1 (containing the GAR) in the cells. The results imply that PI3K delta inhibitors might be therapeutic agents for EBV positive lymphomas.

The experiments are of a high standard and the data are presented clearly. The topic is important and would be of wide interest, suitable for Nature Communications.

There are two main problems that should be addressed:

1. In Figs 1-4, the effect of GAR is studied using a GAR expression plasmid. The results may all be due to over-expression relative to the level of GAR that would normally be present through EBNA1 in a cell containing EBV. EBNA1 expression plasmids with and without the GAR region are readily available in the EBV research community in vectors which replicate as plasmids in mammalian cell lines and thus give physiological expression levels of EBNA1. For example Bill Sugden has published vectors of this type, including p205 which lacks the GAR region in its EBNA1. Authors should test whether this type of EBNA1 expression specifically including the GAR would give the regulation of PI3Kdelta, E2F1 and c-MYC they have observed.

2. The growth inhibitory effects of the PI3Kdelta inhibitor in Fig 5 are clear but they have not been related directly to the presence of EBNA1 (and thus GAR) in the cells. Various different lymphoma and LCL lines with and without EBV are studied but they doubtless all contain a variety of mutations of cell genes that may affect sensitivity to the PI3Kdelta inhibitor. Authors should test the role of EBNA1 and the GAR directly, determining whether introduction of physiological levels of EBNA1 (with or without GAR, as indicated above) into an EBV negative lymphoma cell line causes sensitivity to PI3Kdelta inhibitor when GAR is present.

Reviewer #2 (Remarks to the Author):

In this manuscript, Gnanasundram et al investigated the translational features of GAR repeat sequence derived from EBNA1 and the subsequent oncogenic implications. The authors reported that presence of GAR in the coding region suppressed its own translation, but triggered c-myc induction at the transcription level. The authors then focused on E2F1, a known transcription factor for c-myc, and found translational upregulation of E2F1 as a result of the presence of GAR elements. In searching for the putative mechanism, the authors reported that the upstream signaling pathway mediated by PI3K δ is responsible for E2F1 translation. Consequently, chemical inhibitors of PI3K δ exhibit anti-cancer effects, especially for EBV-carrying cancers.

The unique feature of GAR repeat sequence in translational regulation is intriguing and the elucidation of the communication from cis-sequence elements to trans-acting factors is of significance. Although the entire manuscript was well-written and the results clearly presented, it unfortunately lacks several key mechanisms. For instance, it is unclear how PI3K δ is specifically activated by GAR sequence. How does PI3K δ trigger E2F1 translation? Does this feedback loop only serve to restore ribosomal activity? Fully addressing any of these questions will be of important.

Leaving these fundamental gaps significantly weakens the entire story.

Main concern:

1. I have a hard time in understanding "translation stress". Presence of GAR repeat sequence suppresses its own translation, but it does not necessarily mean that the global translation is affected in the same way (in fact could be in an opposite way). It is intriguingly mysterious how such a small population of "stressed" ribosomes leads to global response. To address this issue, it might be constructive to look into the behavior of ribosomes translating the GAR sequence per se.
2. Page 7, to identify the GAR-dependent factor, how did the authors focus on E2F1 only? What is the priori knowledge about the connection between E2F1 and c-myc? A better description of the rationale is needed.
3. Page 9, a similar increase of E2F1 was observed in the presence of MG132 (Fig. 3f). Why did the authors claim that E2F1 not bound to pRb has a high turnover rate? In Fig 3f, the accumulation of E2F1 seems to be similar in cells with or without GAR.
4. Page 11 and Figure 5, the authors tested a series of inhibitors targeting upstream signaling pathways. What are the controls? PI3K and mTOR inhibitors are known to suppress protein synthesis, which did not seem to be obvious in Figure 5a. In addition, it is unlikely that E2F1 is the only downstream target of PI3K δ . What happens to global translation? Figure 4c suffers from the same issue (lack of cellular control).
5. Page 14, PI3K inhibitors act upstream of AKT and mTOR, but it is a bit odd to claim that mTOR inhibitors are not broad. In fact, mTOR also acts on translation elongation (Faller, et al. Nature 2015; 517).
6. Figure 4, the translational regulation of E2K1 by GAR is potentially interesting. Based on the strong induction of $\Delta(1-324)$, the authors speculated the release of a negative acting factor. This is worth further pursuing by conducting additional experiments (e.g., mRNA pulldown coupled with proteomics). Likewise, identification of PI3K δ substrates that control E2F1 synthesis will be equally important. Otherwise, the entire study does not seem to be complete.

Reviewer #3 (Remarks to the Author):

Review for Gnanasundram et al

In this paper, Gnanasundram et al provide results supporting a mechanism by which GAR of EBNA1 stimulates PI3K δ -mediated induction of E2F mRNA translation leading to c-Myc induction and consequent stimulation of proliferation. Overall, this study highlights a previously largely unappreciated mechanism that might develop better strategies for treating EBV-carrying cancers. Notwithstanding that I appreciated the findings, I thought that criticisms as far as the accuracy and clarity of figures and inclusion of appropriate controls are warranted. My specific comments are provided below:

Major Concerns:

1. Fig 1a: Authors should clearly provide a heading for colony formation assay. This data should be quantified and colony formation should be checked at two independent time points (e.g. day 15 and day 21). Based on the authors, these colonies were stained after 15 days (line 548) whereas in Methods it says 2-3 weeks (Line 391). Authors should amend the method section and put the specific number of days based on their assay.

2. Fig 1b: Authors should indicate the % of serum in their growth conditions and specify the cell line used. Westerns indicate GAr levels are slightly increased in serum starved cells compared to normal or confluent cells. Authors should comment on this trend in the discussion.
 3. Fig 1c and 1e: Authors should make a schematic or indicate 5' or 3' labeling their constructs (e.g. 3'GAR-p535') for visual clarity.
 4. Fig 1d: How was Western blot quantified?
 5. Fig 1f: Authors should also include mRNA levels of additional Myc-target genes (at least 3-5).
 6. Fig 2c: Authors should label (-107/+34) as "Wild-type" in this figure.
 7. Fig2d: Authors should include a positive control in their ChIP analysis.
 8. Fig 2e: Authors should include the mut1 data as well.
 9. Fig 3 (qPCR panels): Are the genes of interest normalized over the house keeping gene? If so, this should be stated in the legends. Authors should also include genes whose expression remains unaffected to show that these effects are specific to their genes.
 10. Fig 4c: Data are presented in inappropriate way. RT-qPCRs should be ran in each fraction and results should be expressed as a % of mRNA in each fraction compared to total mRNA on the gradient showing distribution of E2F1 mRNA levels in both EV and GAr cells. Absorbance profiles for both cells should also be shown. Additional genes that do not shift should also be included to show that the effects of GAr on translation are selective and not caused by the global changes in protein synthesis.
 11. Fig 5a and 5b: Loading should be included for every single blot
 12. Fig 5d is of insufficient quality for publication. All blots are overexposed.
 13. Fig 5f: What about EV cell as a control?
 14. Fig 5: Authors should include the name of the cell type used in every figure subpanel to avoid any confusion.
 15. Methods and legends require more details. Statistical tests used should be specified in each legend for the figures where statistics were used. Authors used GAPDH to normalize at least some of qPCR. It should be specified whether such normalization was used for all qPCRs in each legend.
- Minor Concerns:
1. Authors should include a brief section on cell culture in the main Methods section and guide the readers to supplementary figure.
 2. A brief section on statistical analyses should be included in the Methods section mentioning the tests used to determine significance.
 3. Fig 1d: c-Myc is well-known for promoting ribosome biogenesis. Authors should asses the levels of pre-rRNA by qPCR between EV and GAr cells.

Hope that our comments will hold sufficient pathos with the authors

I/ Topisirovic

Response to reviewers' comments:

Reviewer #1 (Remarks to the Author):

The EBV EBNA1 protein sequence contains a region (GAR) composed of repeated gly and ala amino acids. GAR is not required for the viral DNA replication functions of EBNA1 but is known to play a role in stabilizing the EBNA1 protein and in regulating the translation of EBNA1.

There are two main novel features in this paper. The first is the effect of the GAR on PI3Kdelta and the mechanism of how this induces E2F1 activity and c-MYC (Figs 1-5). The second is the growth inhibitory effect of a PI3K delta inhibitor on EBV positive lymphoma cells, linked to the presence of EBNA1 (containing the GAR) in the cells. The results imply that PI3K delta inhibitors might be therapeutic agents for EBV positive lymphomas.

The experiments are of a high standard and the data are presented clearly. The topic is important and would be of wide interest, suitable for Nature Communications.

We are of course happy to see this positive response and for the good suggestions.

There are two main problems that should be addressed:

1. In Figs 1-4, the effect of GAR is studied using a GAR expression plasmid. The results may all be due to over-expression relative to the level of GAR that would normally be present through EBNA1 in a cell containing EBV. EBNA1 expression plasmids with and without the GAR region are readily available in the EBV research community in vectors which replicate as plasmids in mammalian cell lines and thus give physiological expression levels of EBNA1. For example Bill Sugden has published vectors of this type, including p205 which lacks the GAR region in its EBNA1. Authors should test whether this type of EBNA1 expression specifically including the GAR would give the regulation of PI3Kdelta, E2F1 and c-MYC they have observed.

We have now examined E2F1 and c-Myc induction using EBNA1 WT and EBNA1 ΔGAR constructs and the results are now included in new figures 1g and 2e. These results are well in accordance with our earlier observation that the induction of E2F1 is GAR-dependent. Please also see the results of the low EBNA1 expressing lymphoma cells derived from the transgenic animals (Fig. 6).

2. The growth inhibitory effects of the PI3Kdelta inhibitor in Fig 5 are clear but they have not been related directly to the presence of EBNA1 (and thus GAR) in the cells. Various different

lymphoma and LCL lines with and without EBV are studied but they doubtless all contain a variety of mutations of cell genes that may affect sensitivity to the PI3Kdelta inhibitor. Authors should test the role of EBNA1 and the GAR directly, determining whether introduction of physiological levels of EBNA1 (with or without GAR, as indicated above) into an EBV negative lymphoma cell line causes sensitivity to PI3Kdelta inhibitor when GAR is present.

Figure 5 relates to GAR-mediated activation of E2F1 via PI3Kdelta signaling pathway so we assume the reviewer refers to figure 6? Figure 6 shows data from cells derived from two transgenic mouse lines with, or without, EBNA1. The former is sensitive to CAL-101 treatment but the second is not. The data in 6a, b, c and d are from primary transgenic tumours and are representative from at least 5 different tumours. Each time the experiments are conducted new tumour cells are harvested. As the reviewer points out, tumours will have a different mutation profile, but they all have the common feature of EBNA1 expression. All tumour cells from the EBNA1-expressing transgenic mouse line respond to CAL-101 treatment and the non-EBNA1 do not. Of note, EBNA1 expression levels in these mouse tumours is lower than most LCLs (ie is at physiological levels) and this is published. In figures 6f we show that CAL-101 treatment reduces E2F1 expression in two BL cells lines independent on EBV. This is in line with figure 5f where we show that PI3Kdelta controls E2F1 also in non-GAR expressing cells. However, figure 6g shows that c-myc expression is not affected by CAL-101 in BL cells Raji carrying a translocated myc, only in the EBV-transformed B-cell line B95-8. Both Raji and B95-8 cells show a reduced E2F1 expression following CAL-101 treatment. These results offer an explanation to the previous enigma why there is a reverse relationship between lymphoma phenotype and EBNA1 expression in transgenic two independent mice models. Hence, it is not the EBNA1 protein that is oncogenic –it is its translation inhibitor capacity.

Together with the numerous experiments (plus the new figures 1f and 2f) showing that expressions of GAR results in a cell type-independent PI3Kdelta-mediated increase in E2F1 expression, we believe it is unlikely that the response to CAL-101 in the EBNA1 transgenic lymphoma cells could be due to non-specific effects.

Reviewer #2 (Remarks to the Author):

In this manuscript, Gnanasundram et al investigated the translational features of GAR repeat sequence derived from EBNA1 and the subsequent oncogenic implications. The authors reported that presence of GAR in the coding region suppressed its own translation, but triggered c-myc induction at the transcription level. The authors then focused on E2F1, a known transcription factor for c-myc, and found translational upregulation of E2F1 as a result of the

presence of GAR elements. In searching for the putative mechanism, the authors reported that the upstream signaling pathway mediated by PI3K δ is responsible for E2F1 translation. Consequently, chemical inhibitors of PI3K δ exhibit anti-cancer effects, especially for EBV-carrying cancers.

The unique feature of GAR repeat sequence in translational regulation is intriguing and the elucidation of the communication from cis-sequence elements to trans-acting factors is of significance. Although the entire manuscript was well-written and the results clearly presented, it unfortunately lacks several key mechanisms. For instance, it is unclear how PI3K δ is specifically activated by GAR sequence. How does PI3K δ trigger E2F1 translation? Does this feedback loop only serve to restore ribosomal activity? Fully addressing any of these questions will be of important. Leaving these fundamental gaps significantly weakens the entire story.

Thank you for these encouraging comments. Of course one would like to know more and we agree that we still do not know how PI3Kdelta is activated and this is a priority for our continued studies. However, this is not likely to be simple and will require extensive studies. Down-stream of PI3Kdelta, on the other hand, we are hopeful that we soon will identify key factors. But first we need to publish this novel pathway and hopefully this will stimulate others to join. It is also possible that these data (see above and below) can have clinical implications for EBV-carrying cancers and as such they should be published asap.

The point if this feedback only serves to restore ribosomal activity is interesting. This has not been directly addressed, except for treating the cells with the anti-myc compound, but it is likely that E2F1 induction per se will have growth promoting effects via, for example, the induction of cyclins. Growth and proliferation are closely interlinked.

Viruses explore existing pathways and in figure 5 we show that E2F1 expression is P13Kdelta dependent also in GAR-negative cells. Hence, this pathway is active in proliferating cell lines with a high level of protein synthesis.

Main concern:

1. I have a hard time in understanding "translation stress". Presence of GAR repeat sequence suppresses its own translation, but it does not necessarily mean that the global translation is affected in the same way (in fact could be in an opposite way). It is intriguingly mysterious how such a small population of "stressed" ribosomes leads to global response. To address this issue,

it might be constructive to look into the behavior of ribosomes translating the GAR sequence per se.

We have used the term “translation stress” as the GAR inhibits translation of its own message. We have no evidence the GAR affects translation in trans except for the induction of E2F1 mRNA translation. GAR-dependent translation inhibition is unlikely to affect global translation as it is in the viral strategy to affect as little as possible of the cell but we do not rule out other mRNAs. The signal from the stressed ribosome is enhanced via the PI3Kdelta signaling pathway. The effect of the GAR on its own polysome has been addressed by our team back in 2009 but it is hard from these studies to say exactly what is going on, except that the mechanism involves initiation. In an ongoing parallel study we have direct evidence that it is on the translation initiation (not published). We are slowly starting to patch together the different effects of the GAR and the underlying molecular mechanisms of its action.

2. Page 7, to identify the GAR-dependent factor, how did the authors focus on E2F1 only? What is the priori knowledge about the connection between E2F1 and c-myc? A better description of the rationale is needed.

It is actually a long and winding story starting with the observation that the GAR induces cell growth and from there one thing led to the other. There is long standing link between EBNA1 and c-myc in Burkitts Lymphoma but the link between EBNA1 and c-myc in non-cancer cells has not been investigated. Plus we know about the reverse phenotype between EBNA1 transgenic mice and a lymphoma phenotype. From the induction of c- myc it is logic to look at E2F1. The PI3Kdelta was surprising. It is, in fact, surprising that this pathway has not been discovered previously but most studies on EBNA1 are conducted with a GAR-deleted EBNA1 construct as the GAR suppresses translation and causes aggregates.

3. Page 9, a similar increase of E2F1 was observed in the presence of MG132 (Fig. 3f). Why did the authors claim that E2F1 not bound to pRb has a high turnover rate? In Fig 3f, the accumulation of E2F1 seems to be similar in cells with or without GAR.

E2F1 is kept inactive/stable by pRb but free active E2F1 has a high turn-over rate. Our data show that when we overexpress pRB we observe an increase in E2F1 levels in GAR-expressing cells, suggesting that the GAR is inducing a pool of free E2F1 that is “rescued” by pRb. Overexpression of E7 has less effect on GAR-induced E2F1 levels, in line with the idea that most E2F1 in cells that do not express GAR are bound to pRb. Hence, both MG132 and pRb will “rescue” this pool of free active E2F1 induced by the GAR. We have better clarified this in the text.

4. Page 11 and Figure 5, the authors tested a series of inhibitors targeting upstream signaling pathways. What are the controls?

As suggested by the reviewer, we have repeated this experiment again with appropriate loading controls (please see new figures 5a and 5b.)

PI3K and mTOR inhibitors are known to suppress protein synthesis, which did not seem to be obvious in Figure 5a.

Our results (please see figure 5a panels) indicate that both PI3K and mTOR inhibitors suppresses E2F1 synthesis under normal conditions, however mTOR inhibitor rapamycin fails to suppress E2F1 synthesis under GAR induced translation stress. We also assessed the level p-4EBP1 which was significantly reduced in both cases (please see new figure 5a).

In addition, it is unlikely that E2F1 is the only downstream target of PI3K δ . What happens to global translation?

This is a good point and, indeed, it is unlikely that E2F1 is the lone candidate for this response. But it is worth pointing out that other E2Fs, despite sequence similarities, are not affected; indicating a level of specificity. However, we have previously shown that global translation is not affected by the GAR, in line with the data here showing here that Akt not mTOR pathways are not affected. As mentioned above, we are hopeful that we soon have a better idea of what lies down-stream of PI3Kdelta and together with fine-mapping of the RNA-response element we can more easily assess what other mRNAs might be affected. (please see also Supplementary Fig. 5)

Figure 4c suffers from the same issue (lack of cellular control).

The earlier figure was on analysis of E2F1 mRNA on free pool, light polysome (LP) and heavy polysome fractions (HP), in all the case E2F1 mRNA values were normalized from cellular control (actin). Nevertheless, as suggested by other reviewer, we have repeated this experiment by analyzing individual fractions normalized with cellular control (Please see new figure 4c).

5. Page 14, PI3K inhibitors act upstream of AKT and mTOR, but it is a bit odd to claim that mTOR inhibitors are not broad. In fact, mTOR also acts on translation elongation (Faller, et al. Nature 2015; 517).

We were not clear. We do not claim that mTOR inhibitors are not broad, we meant that broad inhibitors of PI3K catalytic subunits, such as wortmannin and LY294002 can efficiently block

GAr-dependent induction of E2F1, whereas inhibitors of AKT or mTOR that control general cap-dependent translation have little, or no effect on this.

6. Figure 4, the translational regulation of E2K1 by GAr is potentially interesting. Based on the strong induction of Δ (1-324), the authors speculated the release of a negative acting factor. This is worth further pursuing by conducting additional experiments (e.g., mRNA pulldown coupled with proteomics). Likewise, identification of PI3K δ substrates that control E2F1 synthesis will be equally important. Otherwise, the entire study does not seem to be complete.

Thank you for the suggestions, as we mentioned above we are currently working on identifying the substrates of PI3K δ relevant to E2F1 activation which would help in addressing its molecular mechanism. We are making progress and we have a potential candidate that is, however, not a classic translation factors so addressing the molecular mechanism will take more time and we believe such a study will make a good follow up paper.

Reviewer #3 (Remarks to the Author):

Review for Gnanasundram et al

In this paper, Gnanasundram et al provide results supporting a mechanism by which GAr of EBNA1 stimulates PI3Kdelta-mediated induction of E2F mRNA translation leading to c-Myc induction and consequent stimulation of proliferation. Overall, this study highlights a previously largely unappreciated mechanism that might develop better strategies for treating EBV-carrying cancers. Notwithstanding that I appreciated the findings; I thought that criticisms as far as the accuracy and clarity of figures and inclusion of appropriate controls are warranted. My specific comments are provided below:

We are of course happy for this positive response and we hope the new version will be ok.

Major Concerns:

1. Fig 1a: Authors should clearly provide a heading for colony formation assay. This data should be quantified and colony formation should be checked at two independent time points (e.g. day 15 and day 21). Based on the authors, these colonies were stained after 15 days (line 548) whereas in Methods it says 2-3 weeks (Line 391). Authors should amend the method section and put the specific number of days based on their assay.

The colonies were stained after 15 days. We will correct the methods section. We have now included the quantified data from three experiments in main figure 1 and one illustrative image in supplementary fig 1a. Once the E2F1-Myc pathway and the induction of S-phase is shown the colony assays becomes less important.

2. Fig 1b: Authors should indicate the % of serum in their growth conditions and specify the cell line used. Westerns indicate GAR levels are slightly increased in serum starved cells compared to normal or confluent cells. Authors should comment on this trend in the discussion.

H1299 cell line was grown under normal conditions 10% of foetal bovine serum. 0.1% serum was used for starvation conditions. We agree that the level of GAR is slightly increase in starving conditions when compared to normal, but this is not the always case, whereas the increase in cells in S phase is always observed.

3. Fig 1c and 1e: Authors should make a schematic or indicate 5' or 3' labeling their constructs (e.g. 3'³GAr-p535') for visual clarity.

As suggested, we have labelled constructs accordingly.

4. Fig 1d: How was Western blot quantified?

Blots were quantified based on densitometry analysis of c-Myc bands using image j software normalized against actin levels.

5. Fig 1f: Authors should also include mRNA levels of additional Myc-target genes (at least 3-5).

We have included other 5 Myc-target genes including Pol I (please see new figure 1h).

6. Fig 2c: Authors should label (-107/+34) as "Wild-type" in this figure.

Corrected

7. Fig2d: Authors should include a positive control in their ChIP analysis.

8. Fig 2e: Authors should include the mut1 data as well.

We have now fused the data from previous two figures into one (new figure 2d). This saves space and at the same time shows mut1, mut2 and CHIP control data. The DNA CHIP data with endogenous c-myc promoter was included in supplementary figure 4a.

9. Fig 3 (qPCR panels): Are the genes of interest normalized over the house keeping gene? If so, this should be stated in the legends. Authors should also include genes whose expression remains unaffected to show that these effects are specific to their genes.

Yes in all case it was either normalized against actin or GAPDH. We have now clarified this better in each figure legends.

10. Fig 4c: Data are presented in inappropriate way. RT-qPCRs should be ran in each fraction and results should be expressed as a % of mRNA in each fraction compared to total mRNA on the gradient showing distribution of E2F1 mRNA levels in both EV and GAr cells. Absorbance profiles for both cells should also be shown. Additional genes that do not shift should also be included to show that the effects of GAr on translation are selective and not caused by the global changes in protein synthesis.

The earlier figure showed E2F1 mRNA levels on pooled sucrose fractions such as i) free pool, ii) light polysome (LP) and iii) heavy polysome fractions (HP). The E2F1 mRNA levels were normalized from cellular control (actin) and the ratio of normalized E2F1 mRNA levels between EV and GAr was plotted. Nevertheless, as suggested by the reviewer, we have repeated this experiment by analyzing E2F1 mRNA levels in individual sucrose fractions normalized with cellular control (Please see new figure 4c). Also we tested other mRNA levels (p21 and Bip) in the same fractions which does not show any change in levels in polysomes (Supplementary Fig. 5).

11. Fig 5a and 5b: Loading should be included for every single blot

These experiments were repeated accordingly and actin was included as a loading control. Please see new figures 5a and 5b.

12. Fig 5d is of insufficient quality for publication. All blots are overexposed.

The experiments were repeated and please see new figure 5d.

13. Fig 5f: What about EV cell as a control?

EV control is now included (new figure 5f).

14. Fig 5: Authors should include the name of the cell type used in every figure subpanel to avoid any confusion.

All panels in figure 5 belong to H1299 cells. This is now mentioned in the figure legend.

15. Methods and legends require more details. Statistical tests used should be specified in each legend for the figures where statistics were used. Authors used GAPDH to normalize at least some of qPCR. It should be specified whether such normalization was used for all qPCRs in each legend.

We have now better described the controls used for normalization and statistical analysis in each figure legends.

Minor Concerns:

1. Authors should include a brief section on cell culture in the main Methods section and guide the readers to supplementary figure.

We have modified Methods according to the suggestions. Please refer to main Methods section.

2. A brief section on statistical analyses should be included in the Methods section mentioning the tests used to determine significance.

We have now included the statistical analysis used.

3. Fig 1d: c-Myc is well-known for promoting ribosome biogenesis. Authors should assess the levels of pre-rRNA by qPCR between EV and GAR cells.

As suggested, we analyzed the pre-rRNA and mRNA levels of two ribosomal proteins (please refer to figure 1h). Under GAR induction pre-rRNA levels were significantly increased.

Reviewers' comments:

Reviewer #1 (Remarks to the Author):

Two main problems were identified in the previous review. These have not been addressed adequately in the revised manuscript.

1. Were the results an over-expression phenomenon that would not be replicated by expression of EBNA1 at the level normally found in cells infected by EBV?

Authors have now provided helpful data in Fig 1g and 2e to support their model but they have not shown the level of EBNA1 protein in the H1299 cells relative to an EBV infected cell line. So it may still be an over-expression phenomenon. They need a western blot of the EBNA1 and delta GAR EBNA1 in the H1299 cells compared to an EBV infected cell line, also showing a loading control (eg actin or GAPDH).

2. The effects of the inhibitors were not linked directly to the expression of EBNA1 GAR.

This point has not been addressed with any further experimental data. Why don't they compare the effects of the inhibitors on the H1299 cells with EBNA1 or the delta GAR EBNA1 or set up the same comparison in the BL41 cell line used in Fig 6f?

Reviewer #2 (Remarks to the Author):

The authors have addressed my previous concerns and as a result the manuscript has been improved. Although I would have hoped to see more mechanistic connection between GAR translation and PI3K signaling pathway, the current manuscript has already contained a large amount of data. I therefore support its publication in Nature Communications.

Reviewer #3 (Remarks to the Author):

I found that authors responded satisfactory to my queries satisfactory, and I have no further concerns regarding their manuscript being published. Congrats.

I/ Topisirovic

Please see below our detailed response in *italics*.

Response to reviewers' comments:

Reviewer #1 (Remarks to the Author):

Two main problems were identified in the previous review. These have not been addressed adequately in the revised manuscript.

1. Were the results an over-expression phenomenon that would not be replicated by expression of EBNA1 at the level normally found in cells infected by EBV? Authors have now provided helpful data in Fig 1g and 2e to support their model but they have not shown the level of EBNA1 protein in the H1299 cells relative to an EBV infected cell line. So it may still be an over-expression phenomenon. They need a western blot of the EBNA1 and delta GAR EBNA1 in the H1299 cells compared to an EBV infected cell line, also showing a loading control (eg actin or GAPDH).

Please see new Supplementary Fig. 4f. We have now compared EBNA1 expression levels between transfected H1299 cells and the endogenous Burkitt's lymphoma Raji cells. Together with results in Figure 6 (results from the EBV infected cells and low EBNA1 expressing lymphoma cells derived from the transgenic animals) it shows that activation of E2F1 and c-Myc by GAR is unlikely due to a transfection-related phenomenon.

2. The effects of the inhibitors were not linked directly to the expression of EBNA1 GAR. This point has not been addressed with any further experimental data. Why don't they compare the effects of the inhibitors on the H1299 cells with EBNA1 or the delta GAR EBNA1 or set up the same comparison in the BL41 cell line used in Fig 6f?

Please see new Supplementary Fig 7d. We treated H1299 cells expressing vector control (EV), EBNA1 WT or EBNA1 ΔGAR with the PI3Kδ inhibitor CAL-101. The graph is from three independent experiments and shows that the effect of CAL-101 on E2F1 expression in control cells (EV) and in cells expressing the EBNA1 ΔGAR is similar and consistent with the suppression of E2F1 observed in cells treated with siRNA against PI3Kδ. The strongest effect of CAL-101 was observed in cells expressing EBNA1, which is in line with the induction of E2F1 observed in various cells expressing the GAR alone, or GAR fused to different open reading frames.

Reviewer #2 (Remarks to the Author):

The authors have addressed my previous concerns and as a result the manuscript has been improved. Although I would have hoped to see more mechanistic connection between GAR translation and PI3K signaling pathway, the current manuscript has already contained a large amount of data. I therefore support its publication in Nature Communications.

We are, of course, happy to see this positive response.

Reviewer #3 (Remarks to the Author):

I found that authors responded satisfactory to my queries satisfactory, and I have no further concerns regarding their manuscript being published. Congrats.

Thank you for the kind note and for your help with the MS.

REVIEWERS' COMMENTS:

Reviewer #1 (Remarks to the Author):

Authors have now responded appropriately to the comments, including additional data to address the specific points raised.